# Do Females in a Unisexual-Bisexual Species Complex Differ in Their Behavioral Syndromes and Cortisol Production?

**DOI:** 10.3390/biology10030186

**Published:** 2021-03-03

**Authors:** James J. Muraco, Dillon J. Monroe, Andrea S. Aspbury, Caitlin R. Gabor

**Affiliations:** 1Population and Conservation Biology Group, Department of Biology, Texas State University, San Marcos, TX 78666, USA; muracoj@gmail.com (J.J.M.J.); djm261@txstate.edu (D.J.M.); Aspbury@txstate.edu (A.S.A.); 2The Xiphophorus Genetic Stock Center, Texas State University, San Marcos, TX 78666, USA

**Keywords:** glucocorticoids, mate choice, personality-traits, *Poecilia latipinna*, *Poecilia formosa*

## Abstract

**Simple Summary:**

In many species, including humans, individuals in a population have personalities: collections of correlated behaviors that are consistent across different environments (i.e., mating, eating). Personalities are affected by competitors for food or mates and the hormones produced by individuals). Competitors can include other individuals of the same species or closely related species. The all-female, Amazon molly is a hybrid species, and needs to coexist with one of its bisexual (males and females), parent species, to reproduce. One parent species of the Amazon molly is the sailfin molly. Female sailfin and Amazon mollies compete for access to males for mating and food which could affect the personalities of individuals of each species. We found that both species have similar personalities consisting of a correlation between exploration and activity. We did not detect a relationship between a stress response hormone, cortisol, and individual personality. However, the all-female Amazons had higher cortisol release rates than sailfins. Personalities may be similar due to genetic constraints that link these behaviors, and might benefit Amazons if this causes male sailfin mollies to mismate with them. However, the differences in cortisol release rates may be a useful mate identification cue for males to offset such mating mistakes.

**Abstract:**

Studies of suites of correlated behavioral traits (i.e., behavioral syndromes) aid in understanding the adaptive importance of behavioral evolution. Behavioral syndromes may be evolutionarily constrained, preventing behaviors from evolving independently, or they may be an adaptive result of selection on the correlation itself. We tested these hypotheses by characterizing the behavioral syndromes in two sympatric, closely related species and testing for differences between the species. We studied the unisexual Amazon molly (*Poecilia formosa*) and one of its bisexual, parent species, the sailfin molly (*P. latipinna*). Sympatric female sailfin and Amazon mollies compete for mating which could affect the behavioral syndromes found in each species. We identified a behavioral syndrome between exploration and activity in both species that did not differ between species. Additionally, we explored the relationship between a stress response hormone, cortisol, and behavioral type, and did not detect a relationship. However, *P. formosa* differed from *P. latipinna* in their cortisol release rates. Behavioral syndromes may be constrained in this complex, aiding in mate acquisition for *P. formosa* by virtue of having a similar behavioral type to *P. latipinna*. The difference between the females in cortisol release rates may be a useful mate identification cue for males to offset higher mating mistakes associated with the similar behavioral types.

## 1. Introduction

A framework for studying behavior has been developed, which emphasizes the need to obtain a more holistic view of animal behavior through the examination of suites of correlated traits that are consistent across multiple environmental situations and time [1,2,3,4]. This framework has been termed “behavioral syndromes” and is similar in principle to other terms in the literature such as “coping style” [5,6] and “personality” [7]. Indeed, the hypothalamic-pituitary-inter-renal (HPI) axis is associated with certain coping styles across environments indicating that hormones mediate consistent behavioral reactions, and this creates pleiotropic action on the behavior and physiology of animals. Yet, because complex patterns underly the correlation between behavior and physiology some studies do not find such correlations [8,9].

The behavioral syndrome framework recognizes that there is often consistent, individual variation in behavior both within and across environmental situations. There are competing hypotheses explaining the ecological and evolutionary consequences of behavioral syndromes at the population or species level [2,10,11,12]. For instance, the ‘constraint hypothesis’ suggests that behavioral traits become genetically correlated (constituting the behavioral syndrome) through pleiotropy, resulting in tight genetic linkages, such that selection on one trait results in selection on the correlated trait(s) as a byproduct [13,14,15,16]. Therefore, groups of correlated behaviors may act as a constraint on the independent evolution of individual behaviors [17]. The ‘adaptive hypothesis’ in contrast suggests that correlated behavioral traits are adaptations present within populations because of selection on the correlation itself [16,18,19]. Given that a behavioral syndrome is a population-level property [15], these two hypotheses allow researchers to make predictions about the presence and maintenance of behavioral syndromes between populations and even between species [16]. Specifically, the ‘constraint hypothesis’ predicts that behavioral syndromes should be similar across populations because even if selection acts on different behaviors in different populations all correlated behaviors are constrained to evolve together in the same direction. Conversely, the ‘adaptive hypothesis’ suggests that correlated behaviors (or perhaps the lack thereof) should differ between populations, the strength and direction of selection on the correlatation between behaviors may differ among populations [16]. Studies in sticklebacks, *Gasterosteus aculeatus* and a poeciliid fish, *Brachyraphis episcopi* support the ‘adaptive hypothesis’, where there are differences in behavioral syndromes across populations [16,20,21,22]. Yet, a more recent study examining genetic correlations in field crickets (*Gryllus integer*) supports the ‘constraints hypothesis’. Field crickets exhibit conservation of behavioral syndromes at the genetic level, despite among population differences in average behavior [23]. Taken together, these results indicate that it is important to consider both genetic variation among populations along with the selective forces that can generate and maintain behavioral syndromes [23,24].

Numerous studies have examined differences in behavioral syndrome composition between populations [16,20,21,22,25,26] but an overlooked question is: how do syntopic species differ in behavioral syndromes? A useful system for testing the constraint vs. adaptive hypotheses is one where two species are closely related and sympatric, as such a system allows for predictions to be made about similarities (or differences) in behavioral syndromes between species based upon shared genetic variation and shared environments. Further, the relationships between physiology and behavior may explain or play a role in animal personality and behavioral syndromes [2,27,28] indicating the need to understand whether between-individual or within-individual correlations dictate patterns of phenotypic covariation among behavioral and physiological traits.

Consistency between individual differences in behavior can be associated with or mediated by consistent between individual differences in physiology [5,29] although the relationships between physiology and behavior are complex. Response to environmental change is mediated by the HPI axis that results via downstream production of glucocorticoids which affects the energetic state of an animal and aids in coping with environmental challenges. Cortisol production, the main fish glucocorticoid [30], can result in the reallocation of resources from functions related to growth and reproduction to behaviors which are likely to increase immediate survival [30,31,32]. The relationship between cortisol and behavioral type is unclear and appears to differ between species. Thomson et al. [33] did not detect a relationship between the individual stress response, as measured by cortisol production and behavioral type (e.g., bold vs. shy individuals) in rainbow trout, *Oncorhynchus mykiss*. Conversely, Archard et al. [34] discovered a positive relationship between cortisol production and the behavioral type, as indicated by measures of exploration, in a tropical poecilid, *Brachyrhaphis episcopi.* Additionally, Muraco, Aspbury and Gabor [35] found that bold individuals had higher pre-stimulus release rates of cortisol than shy individuals in male sailfin mollies (*Poecilia latipinna*). These results suggest that cortisol may be an important factor in the behavioral type of fish.

Sailfin mollies are part of a species complex that lends itself to explore questions about differences in correlated traits across species. The sailfin molly (*Poecilia latipinna*) and the Atlantic molly (*P. mexicana*), and the unisexual gynogenetic Amazon molly (*P. formosa*) are members of a unisexual-bisexual mating species complex. *Poecilia formosa* is a mostly clonal, all-female species of live-bearing fish that arose via a hybridization event between *P. latipinna* and *P. mexicana* [36]. The genome of *P. formosa* is intermediate between the parental species [37]. Sperm from one of the parent species is required by the Amazon molly to start embryogenesis [36]. As such, *P. formosa* are considered sexual parasites of the parent species. Further, *P. formosa* are more aggressive in a mating context than female *P. latipinna* [38] which may be adaptive, as this behavior is likely to increase the frequency of mates obtained by *P. formosa* [38,39]. Previous research has not revealed differences in behavioral syndromes or correlations among behaviors between Amazon and sailfin molly females, but the study suffered from small sample sizes [40]. Recent studies by Seda et al. [41] and Muraco et al. [35] have found evidence of behavioral syndromes in male *P. latipinna* suggesting that they could also be present in female *P. latipinna* and possibly in *P. formosa* due to their closely shared genetic background.

In this study, we examined whether there were differences in behavioral syndromes between the unisexual *P. formosa* and females of the bisexual *P. latipinna*. The adaptive hypothesis predicts that *P. formosa* and *P. latipinna* would have different syndromes due to different selective pressures. *Poecilia formosa* benefit from behaviors that increase the chance of obtaining any mates (i.e., they only need sperm to start the development of their eggs) rather than from high-quality mates, so we predict that they will have a behavioral syndrome associated with no benefit to mate choice (i.e., active, bold and exploratory), whereas female *P. latipinna* will have a different behavioral syndrome partially driven by benefitting from mating with higher-quality males. Alternatively, given that *P. formosa* is a hybrid between *P. latipinna* and *P. mexicana*, the constraint hypothesis would predict that behavioral syndromes will not differ between species as a result of their shared genetic variation. We also hypothesize that there will be a relationship between cortisol production and behavioral type (specific configuration of the behavioral syndrome expressed by an individual), and that this relationship will differ between species. Although there has been mixed support for a relationship between behavioral type and hormone production [33,34,35], we predict that there will be a relationship between behavioral type and pre and post behavioral trial cortisol release.

## 2. Materials and Methods

### 2.1. Fish Maintenance

We tested wild-caught female *P. latipinna*, as well as *P. formosa*, that were collected from the headwaters of the San Marcos River, Spring Lake, San Marcos, Hays County, TX (29.89 N, 97.82 W) from May 2011–January 2012. This population of *P. latipinna* has been sympatric with *P. formosa* for less than 100 years, as *P. latipinna* were introduced in the 1930s and *P. formosa* were introduced in the 1950s [42]. We maintained fishes in the laboratory at 23–25 °C for a minimum of 30 days before experimentation so females were not likely to drop a brood during testing. We housed females in single-sex, mixed-species aquaria (38 L). All tanks had live java moss in them, rocks on the bottom and a sponge filter. We changed half the water midway in each month and did a ¾ water change at the end of each month. We maintained all fish on a 14:10 L:D cycle and fed them a combination of brine shrimp (*Artemia spp*.; Bio-Marine) and Spirulina Flakes mixed with Aquamax^®^ Fry Starter 200 (PMI Nutrition International). We fed fish once daily between 1630 and 1730 h to control for any hormonal changes due to feeding.

### 2.2. Identification of Behavioral Types

The experimental design of this study follows that of Muraco, Aspbury and Gabor [35], excluding the mate preference component. We tested individual females of each species (*n* = 40 *P. formosa*; *n* = 40 female *P. latipinna*) for activity, boldness, and exploration twice. There was at least 1 day between testing for each individual.

Each trial consisted of placing individual females into an opaque, acclimation chamber (16.5 × 15 × 13 cm) for 5 min within an experimental arena (70 × 40 × 10 cm), that was marked with a grid of 36 total squares (each 7.5 × 8.5 cm). Testing occurred between 0800-1300 h daily. Latency to emerge from the acclimation chamber was measured as a proxy for risk-taking behavior or “boldness”. We quantified boldness as log (max time to exit)—(individual time to exit), where max time to exit totaled 600s. Additionally, we quantified activity as the total time (s) that individuals spent moving in the 5 min after emergence. Exploration was calculated as the total number of grids that an individual entered during the 5 min trial. Individuals that move through more grids are considered more exploratory [6]. We repeated our measures of activity, boldness and exploration for all 80 individual females at the same time of day to obtain measures of short-term behavioral repeatability (or consistency). By measuring traits multiple times we are able to test the assumption that individual expression of suites of correlated behaviors is stable over time [12].

### 2.3. Hormone Sampling 

We obtained pre-trial cortisol release rates via water-borne hormone samples 24 h before behavioral trial exposure. Additionally, we obtained post-trial cortisol release rates via water-borne hormone samples after the second trial exposure (and within 24 ± 1 h of the pre-trial sampling). After obtaining hormone samples the mass and standard length (SL) was measured for each fish. Hormone samples were taken between 0800-1300 h to control for circadian variation in hormone levels [43]. Water-borne hormone collection methods followed Gabor and Grober [44]. Briefly, individual females were placed into a 250 mL beaker filled with 100 mL of conditioned water and then removed after 1 h. Water samples were stored at −20 °C until hormone assays were completed at a later date (Ellis et al. 2004). All subsequent hormone assay procedures followed the methods of Gabor and Grober [44] and Gabor and Contreras [45]. We extracted water-borne hormones using C18 solid phase extraction (SPE) columns (Waters Inc, Milford, MA, USA) with a vacuum manifold. We eluted hormones with 4 mL of 100% HPLC grade methanol and then evaporated the solvent under a stream of nitrogen and resuspended the samples in assay buffer from the enzyme-immunoassay kit (№ 501320, Cayman Chemical Company, Inc). All samples were run in duplicate on 96-well plates and read by a fluorescent plate reader (BioTek Powerwave XS). We multiplied cortisol release rates (pg/mL) by the final suspension volume and the standardized the value by dividing by the standard length of the fish (pg/mm/h). All hormone release rates were natural log transformed.

### 2.4. Statistical Analysis

We performed three tests. First, repeatability of behaviors was measured using mixed-effects models with bootstrapped confidence intervals for each species separately. Second, to detect the presence of a behavioral syndrome among each species we used a multivariate mixed model fitted in a Bayesian framework using the MCMCglmm R package [46]. Models were based on 1,050,000 iterations with a burn-in of 50,000 iterations and thinning of 40, resulting in the use of 25,000 iterations to obtain point estimates and 95% credibility intervals. As exploration was not repeatable, we used the average of the two measures along with natural log transformed activity and boldness measures as dependent variables using a Gaussian residual error distribution. For the repeated observations we included individual as a random effect. Third, we compared pre-trial cortisol release rates between *P. formosa* and *P. latipinna* using Welch’s two sample t-test for unequal variances. Fourth, we also examined the relationship between change in cortisol (post trial—pre-trial measures) and behavioral traits using this method. For the residual and between individual variation, we modeled an unstructured variance-covariance matrix following Santicchia, Wauters, Dantzer, Westrick, Ferrari, Romeo, Palme, Preatoni and Martinoli [9]. Average differences in behavioral traits among species were assessed using separate linear mixed-effects models (“lme4” package). For each model, species was included as a fixed factor and fish ID as a random effect to account for repeated measures. The correlations between each behavior and mass and SL were not statistically significant so they were not included in the model (correlation estimates whose 95% CI excluded zero were considered significant). All statistical analyses were performed using the program R v. 3.6.2 [47].

## 3. Results

Boldness was highly repeatable, and activity was moderately repeatable (Table 1). There was no significant correlation between activity and boldness, or boldness and exploration, and the relationships between each behavior did not differ between species (Table 2a,b). Both *P. latipinna* and *P. formosa* had a behavioral syndrome consisting of activity and average exploration (Table 2c). Pre-trial cortisol release rates were significantly higher in *P. formosa* (Welch’s two sample t-test: t = 3.17, df = 27.1, *p* = 0.004; Figure 1). The change in cortisol release rates was not correlated with boldness or activity in either species but the change in cortisol did differ between species, with *P. latipinna* showing a greater change in cortisol release rates than *P. formosa*. (t = −3.8, df = 29, *p* ≤ 0.001; Figure 1).

## 4. Discussion

The unisexual hybrid Amazon mollies (*P. formosa*) and the bisexual, female sailfin mollies (*P. latipinna*) live in sympatry and share close to 50% of the genome with each other [48]. We found that both species showed an activity—exploration correlation and that the relationship between these behaviors did not significantly differ between the two species. Additionally, we found that boldness and activity measures were repeatable, although the exploration measurement was not. These results provide support for the constraints hypothesis that suggests that behavioral syndromes may be similar across species owing to genetic similarity instead of different selection environments, in this case mate choice. The genetic correlation between behavioral traits is not evolving, so selection in one trait equally affects the other traits. The hybrid origin of *P. formosa* may be the basis for these behavioral similarities. Nonetheless, the two species differ in their pre-trial cortisol release rates and their cortisol response (change in cortisol production) associated with their response to exposure to the novel testing apparatus. Yet, we did not find that the behaviors were related to their cortisol response or their pre-trial cortisol release rates. Thus, the behaviors of these fish may be similar owing to constraints, but cortisol release rates are not linked in a clear pattern to the behaviors of the species, and thus could vary between them.

Similar to two other studies we provide evidence of behavioral syndromes in female *P. latipinna*, and *P. formosa,* as has been found for male *P. latipinna* [35,41]. Additionally, females of both species showed repeatability in boldness and activity (Table 1) indicating that these fish have animal personalities. We originally predicted that being active, bold, and exploratory would increase the chances of obtaining a mate. However, we were unable to detect correlations among boldness behaviors like [40,49]. We argue that our data support the constraints hypothesis given the genetic similarities between these two species. An alternative hypothesis for the lack of differences between the two species is that selective forces have resulted in *P. formosa* mimicking the behavior of *P. latipinna* or via the two species having similar selective pressures for reproductive strategies. Behavior mimicry would be adaptive if it increases the probability of obtaining a mate, perhaps from smaller males that are likely to make mating mistakes due to decreased inspection time [50,51]. Unfortunately, our data does not allow us to differentiate between shared behavioral types due to ancestry/shared genetic variation vs. adaptive response. Nonetheless the more parsimonious hypothesis is that the shared genetics is the basis for the similarity in behavioral traits and not due to selection on female *P. formosa* to mimic the behavior of *P. latipinna* because *P. formosa* reproduce clonally (mate choice copying could also play a role [52]). An important follow up to our study would be to compare behavioral syndrome composition between female *P. latipinna* and *P. mexicana* and *P. formosa*. Additionally, our system allows for explicit testing of the adaptive hypothesis, as geographic variation in behavioral syndrome composition would falsify the constraint hypothesis [16].

The similarity of the behavioral syndromes between *P. formosa* and *P. latipinna* may make it harder for males to differentiate between the two species in a mating context, especially given that the male *P. latipinna* already suffer a conflict in species vs. mate-quality recognition [53]. Foran and Ryan [38] found that *P. formosa* was more aggressive on average than female *P. latipinna*, which might result in a higher frequency of mating via blocking of female *P. latipinna* yet we did not find differences in boldness between the two species. Behavior syndromes are defined as traits that persist across contexts, so testing them alone and then interpreting those results in a mating context is not unreasonable [54,55]. However, we did not explicitly test whether females with a specific behavioral type were more likely to obtain a mating; therefore, an interesting follow-up would be to examine whether there is a relationship between behavioral type and mate acquisition abilities both within and between females of both species in this unisexual-bisexual mating complex. 

Hormones regulate behavior in a variety of ways, and there is increasing evidence that hormones and behavioral syndromes are linked, with particular emphasis on stress hormones [5,56]. Some studies do not reveal associations between physiological stress response and behavioral syndromes [9] or individual behaviors [33]. However, Muraco et al. [35] found that bolder male sailfin mollies had higher pre-trail release rates of cortisol than less bold males. In female sailfin mollies and Amazon mollies we did not find a relationship between any individual behavior and cortisol release rates. Nonetheless, we found species-level differences in cortisol release rate. Amazon mollies had higher pre-trial release rates of cortisol than female sailfin mollies (Figure 1), even though the two species did not differ in boldness traits as predicted. Furthermore, there was a difference in the cortisol response between species, such that female sailfin mollies had a higher cortisol response (post-trial-pre-trial) on average than Amazon mollies (Figure 1). Chronic elevation of glucocorticoids (i.e., stress hormones) can result in a reallocation of resources from growth and reproduction to factors crucial to immediate survival [30]. 

When female *P. latipinna* and *P. formosa* occur in sympatry, they are in direct competition for mattings from the same population of males. However, both species may differ in mating preferences because *P. formosa* benefits from any mating, not just high-quality mating. These factors, in turn, could affect the behavioral syndromes found in each species. In this study, we discovered that female *P. latipinna* and *P. formosa* exhibit similar behavioral syndromes making it possible that *P. formosa* may be equally as choosy as *P. latipinna* as a result of having a similar behavioral type to *P. latipinna*. However, *P. formosa* differ in cortisol release rates and cortisol response than *P. latipinna.* These differences in cortisol release rates may aid males in identifying conspecifics despite similarities in behavioral syndromes. Hormones can mediate bidirectional interactions between males and females during courtship. Indeed, Gabor and Grober [44] found bidirectional interaction between male and female *P. latipinna* 11-ketotestosterone release rates. The elevated release rates of cortisol by *P. formosa* may function as a cue that informs males of species identities.

## Figures and Tables

**Figure 1 biology-10-00186-f001:**
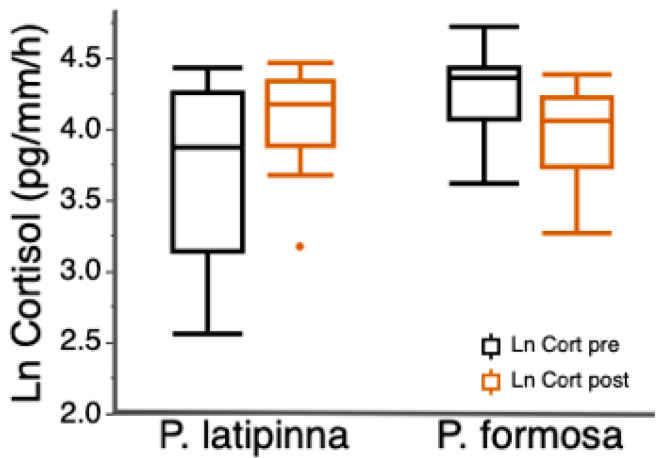
Mean cortisol release rate (ln pg/mm/h) for female *P. latipinna* and *P. formosa* before the behavioral trial (Pre-trial cortisol) and immediately after the behavioral trial (post-trail cortisol). Box plots indicate median, range, and first and third quartiles, dot represents an outlier.

**Table 1 biology-10-00186-t001:** Mixed-effects model results of repeatability of behaviors within and among *Poecilia formosa* and *P. latipinna* (Intraclass correlation coefficient (ICC) values) and comparisons between species for each behavior, both statistics include 95% confidence intervals. Behavioral estimates and 95% confidence intervals for *P. formosa* are the difference between *P. formosa* and *P. latipinna*. Bold values are significant.

	ICC (r)	95% CI	Estimate	95% CI
Activity-among species	**0.12**	**0–0.25**		
*P. latipinna*	**0.01**	**0–0.22**	218.84	201.46–237.20
*P. formosa*	**0.27**	**0–0.39**	−13.05	−36.94–11.23
Exploration-among species	0.00	0–0.18		
*P. latipinna*	0	0–0.20	25.44	21.00–29.58
*P. formosa*	0	0–0.22	0.99	−1.32–3.71
Boldness-among species	**0.28**	**0.06–0.36**		
*P. latipinna*	**0.36**	**0–0.35**	466.92	369.33–565.58
*P. formosa*	**0.34**	**<0.001–0.41**	15.93	−40.89–73.82

**Table 2 biology-10-00186-t002:** Markov chain Monte Carlo multivariate model results for behavior syndromes. Repeated measures of boldness and activity were tested for between and within species syndromes. Non-repeated measures were tested for correlations with boldness and activity for within species syndromes. Bold values are significant.

	Correlation Estimate	95% Credible Intervals
**a. Boldness ~ Activity**		
Among species	0.21	−0.66–0.96
*P. latipinna*	0.28	−0.60–0.99
*P. formosa*	0.05	−0.80–0.90
**b. Boldness ~ Exploration**		
*P. latipinna*	0.20	−0.6–10.96
*P. formosa*	−0.01	−0.83–0.90
**c. Activity ~ Exploration**		
*P. latipinna*	**0.67**	**0.26–1.00**
*P. formosa*	**0.61**	**0.15–0.97**
**d. Activity ~ Change in cortisol**		
*P. latipinna*	−0.44	−1.00–1.00
*P. formosa*	0.74	−1.00–1.00
**e. Boldness ~ Change in cortisol**		
*P. latipinna*	−0.20	−1.00–1.00
*P. formosa*	−0.65	−1.00–1.00

## Data Availability

The raw data supporting the conclusions of this article will be made available by the authors, without undue reservation.

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
