# Peer review of "Do Females in a Unisexual-Bisexual Species Complex Differ in Their Behavioral Syndromes and Cortisol Production?"

_biology, 2021, doi:10.3390/biology10030186_

Round 1
Reviewer 1 Report
This manuscript studies behavioural syndromes in the P. formosa and P. latipinna unisexual-bisexual species complex. The authors question whether the behavioural syndromes between boldness, activity levels and exploration tendency, as well as the behavioural-hormonal syndromes between these behaviours and cortisol release rates, are similar between the two species. Due to the genetic similarity between the two species, similar syndromes are expected if the genetic correlations between these traits are constraining their independent evolution. Alternatively, if selection affects the correlation itself, divergent behavioural syndromes may have evolved since the species’ behavioural ecology is different, especially concerning mating behaviour.
This is a very interesting rationale and the authors cite the appropriate literature on the subject, both conceptual and experimental. Moreover, the authors’ idea of comparing behavioural syndromes between syntopic species seems new and, indeed, pertinent for the type of questions they are asking. However, I am not convinced about the way the authors tested their ideas. But perhaps improvements to the introduction and discussion will be enough to recover the actual relevance of the authors’ work.
Here I describe my major concerns and in the attached manuscripts’ pdf, I describe and provide solutions for some minor concerns:
- Behavioural-hormonal syndromes
In the introduction, the authors are not clear about the conceptual link between behavioural syndromes and hormonal (cortisol) syndromes. They present references that are both in favour and against correlations between behavioural and hormonal traits, which is confusing. The authors should be very clear about how hormones and behaviour may correlate to form specific behavioural-hormonal syndromes, and how can this help them to disentangle between the constraint and the adaptive hypotheses. Some of this information is provided in the discussion, but the introduction also needs to be convincing. In its present version, the introduction provides the misleading idea that the testing of the behavioural syndromes and the cortisol-stress response were independent questions and experiments, which reduces the originality and relevance of the authors work.
- The ecological context in which the experiments were performed
The authors oriented their argument to the mating context: P. latipinna females should be choosier than P. formosa females because the latter do not need the sperm to fertilize the eggs, only to initiate embryogenesis. Therefore, any sperm, whether of high or low quality does not make any difference to P. formosa females. Therefore, the behavioural syndromes of the females of these two species can be expected to be different if there are no genetic correlation constraints that prevent independent evolution between behaviours. This makes sense. However, the experiments were not done in a mating context. They were done in an empty tank, and the latency to emerge from the shelter (boldness), the time spent swimming and the distance explored were measured in this “empty” context. No male or pair of males of different quality (e.g. inside transparent partitions) were present. So, I ask if it is correct to make inferences about female choosiness if their experiments did not place females in this specific condition. Are measures of boldness, activity and exploration in a neutral context generalizable to other contexts? For example, does boldness in a neutral context generally correlate with boldness in any other context? If yes, the authors should cite that literature. Otherwise, they must be more cautious about their extrapolations to the mating context and must specifically acknowledge that they have not done their tests in that context.
- Motivation
Females of both species were tested twice with only 1 hour of rest between trials. Is this the rest time normally used in personality tests? If yes, please, cite the relevant literature. If not, this raises issues about the females’ motivation to explore the test tank after 1h, since they had already explored it before. Could this lack of motivation explain the non-significant result for exploration (non-significant repeatability and non-significant correlation with the other behaviours)? This must be discussed in the manuscript, as may it represent a real limitation of the study.

Author Response
Behavioural-hormonal syndromes
In the introduction, the authors are not clear about the conceptual link between behavioural syndromes and hormonal (cortisol) syndromes. They present references that are both in favour and against correlations between behavioural and hormonal traits, which is confusing. The authors should be very clear about how hormones and behaviour may correlate to form specific behavioural-hormonal syndromes, and how can this help them to disentangle between the constraint and the adaptive hypotheses. Some of this information is provided in the discussion, but the introduction also needs to be convincing. In its present version, the introduction provides the misleading idea that the testing of the behavioural syndromes and the cortisol-stress response were independent questions and experiments, which reduces the originality and relevance of the authors work.
We are not trying to make the argument that behavior may form a behavioral hormone syndrome, rather, we are focusing at the level of the individual hormone release rates to see if there are relationships between behavioral types (the specific configuration of the behavioral syndrome that is expressed by an individual) and hormone release rates. We found similar patterns for male sailfin mollies and this has also been found in another livebearing fish. We have tried to link these concepts together better with a few extra sentences through the paragraphs of the introduction.
The ecological context in which the experiments were performed
The authors oriented their argument to the mating context: P. latipinna females should be choosier than P. formosa females because the latter do not need the sperm to fertilize the eggs, only to initiate embryogenesis. Therefore, any sperm, whether of high or low quality does not make any difference to P. formosa females. Therefore, the behavioural syndromes of the females of these two species can be expected to be different if there are no genetic correlation constraints that prevent independent evolution between behaviours. This makes sense. However, the experiments were not done in a mating context. They were done in an empty tank, and the latency to emerge from the shelter (boldness), the time spent swimming and the distance explored were measured in this “empty” context. No male or pair of males of different quality (e.g. inside transparent partitions) were present. So, I ask if it is correct to make inferences about female choosiness if their experiments did not place females in this specific condition. Are measures of boldness, activity and exploration in a neutral context generalizable to other contexts? For example, does boldness in a neutral context generally correlate with boldness in any other context? If yes, the authors should cite that literature. Otherwise, they must be more cautious about their extrapolations to the mating context and must specifically acknowledge that they have not done their tests in that context.
We understand that we did not measure the behavior of these fish in the presence of a male, but if we had we could not have tested their behaviors repeatedly because each male would interact differently with the females which would in turn complicate our ability to analyze their behaviors. Further, behavior syndromes are defined as traits that persist across contexts, so testing them alone and then interpreting those results in a mating context is not unreasonable. In a study by Castanheira et al. 2013 they explored this specific question and found that fish tested alone and in groups had cross context relationships between risk taking tests and feeding recovery in fish. (https://journals.plos.org/plosone/article?id=10.1371/journal.pone.0062037). Similarly, aggressive behavior has been found to be consistent across contexts (Pruitt, J.N., Jones, T.C. & Riechert, S.E. (2008). Behavioural syndromes and their fitness consequences in a socially polymorphic spider, Anelosimus studiosus. Anim Behav., 76, 871–879.). See review in Sih et al. 2012 (Ecology Letters)
- Motivation
Females of both species were tested twice with only 1 hour of rest between trials. Is this the rest time normally used in personality tests? If yes, please, cite the relevant literature. If not, this raises issues about the females’ motivation to explore the test tank after 1h, since they had already explored it before. Could this lack of motivation explain the non-significant result for exploration (non-significant repeatability and non-significant correlation with the other behaviours)? This must be discussed in the manuscript, as may it represent a real limitation of the study.
Thanks for noting this. This was a mistake. There was 1 day between trails. All trails were at the same time of day for each individual fish too. We have now resolved this in the manuscript.
Reviewer 2 Report
In general I wonder why the authors only investigated corticosterone and not 17ß-Estradiol or both?
17ß-Estradiol influences exploration behaviour aswell as the stress level and reproductive behaviour. Also it is known that the male representatives of a species can react to the Estradiol concentration of the females. ( e.g. T.G.Pottinger et al., 1996, Testosterone, 11-Ketotestosterone, and Estradiol-17β Modify Baseline and Stress-Induced Interrenal and Corticotropic Activity in Trout; General and Comparative Endocrinology; Andréia B. Moraes et al., 2020, Pro-social and anxiolytic-like behavior of a single 24-h 17β-estradiol exposure in adult male zebrafish; Neuroscience Letters https://doi.org/10.1016/j.neulet.2020.135591
Would it be possible to analyse 17ß-Estradiol in the samples and correlate that with the results from CORT? It's known that the authors published a method to analyse 17ß-Estradiol in previous papers.
Why didn’t the authors use the ACTH challenge test to find intraindividual differences in behaviour before and after the treatment and behavior tests. The ACTH challenge test is a common tool for investigations regarding stress answers
Why did the authors use animals caught in the wild? How did the authors exclude any infections in these animals? Wild caught animals are often infected with parasites. In this case cortisol could be secreted in an increased level because of its immunosupressive function.
Line 20 How do the authors define relationship and what type of relationship do they expect? please add elsewhere
Line 21 What do the authors mean by „behavioural type“? please add elsewhere
Line 107-110 Neither the execution nor the results of this study allow for such a far-fetched statement. Please reword.
Line 111 A multitude of environmental factors influence the secretion of cortisol and it is known that water change can trigger mating behavior. Hence, add the following parameters:
- How long did one filter cycle of the water take.
- How often were water changes conducted.
- How much time was between the last water change and behaviour exam.
- Was the aquarium enriched or barren. If it was enriched, how?
Please add the range of the water temperature. Is it the same water temperature as in their natural habitat?Are there natural changes, e.g. night lower than day because of influences of sunlight radiation?
Please add information: are the species nocturnal or diurnal, are there differences in the biological rhythm of the investigated species? usually species, that live in the same habitat and need the same diet, have an ecological niche. for example, do they differ in their active phase? Please describe the differences of the species.
Line 143 At what time of day (which time frame) were the behavioural exams conducted (e.g. only in the morning, only in the evening?) Please add how the authors differentiate between activity and exploration behavior
do you know the circadian rhythm of cortisol in the mentioned species? please add
Were the behavioural exams taken in the same time frame as the hormone sampling (0800h-1300h)?
Line 147 Please describe the hormone assay procedures in more detail. It is tedious to look up references, so please summarise the contents of each reference as usual.
Line 148 Are the authors sure that the methods of reference 44 are fitting, because only sexual hormones were investigated there
Line 169 please add results regarding anxiety and escape behaviour, because they can also be influenced by corticosterone.
results
Figure 1: I don't understand the unit (Ln Cotisol (pg/mm/h)). This unit is used in the hematology for blood sedimentation but not for the concentration of hormones determined via EIAs. The common unit here is pg/mmol or ng/ml, respectively. Please explain the calculation.
Table 2 : of course it could excpected that the animals show less activity/exploration behavior in the second run. The fish know the "new" environment of the test chamber already and explore less. Did you also measure cortisol in the second run? The cortisol secretion should also be lower then. How do the results differ from the first? Please add.
discussion
the authors should first discuss if their hypotheses are fulfilled or not and why or why not.
line 243 please remove "did"
line 254 matings are possible several times in a row with different females
line 261 in the mentioned case I would expect sexual hormones than stress hormones suit better
Please decribe the advantages and disadvantages of wild caught animals in this study
Round 2
Reviewer 1 Report
This is the second time that I have revised this manuscript.
The authors made some changes to the introduction and methods that slightly improved the quality of their manuscript.
Namely, that the same fish were tested in different days, solving the motivation issue and that behaviour types and hormone release rates do not form behavioural-hormone syndromes, but instead that hormone release rates may be part of the underlying mechanism of behavioural syndromes.
For the ecological context in which the experiments were performed, the authors explained that «behaviour syndromes are defined as traits that persist across contexts, so testing them alone and then interpreting those results in a mating context is not unreasonable». This is important information to the readers that the authors, however, did not include in the manuscript. I advise them to do so, including with the references Castanheira et al. (2013) and Pruitt et al. (2008).
In my previous review, I also provided a pdf file describing 65 additional (minor) concerns and providing suggestions to solve most of them. Did the authors receive this file? I am attaching the file again in case the authors and the editor still find it useful.

Author Response
Thank you for your comments and edited PDF. We am very sorry that we didnt see the PDF the first time around. We have now addressed both the comments below and those that were not already address in the pdf.
In response to your comments below
For the ecological context in which the experiments were performed, the authors explained that «behaviour syndromes are defined as traits that persist across contexts, so testing them alone and then interpreting those results in a mating context is not unreasonable». This is important information to the readers that the authors, however, did not include in the manuscript. I advise them to do so, including with the references Castanheira et al. (2013) and Pruitt et al. (2008).
We have added a sentence similar to the suggested sentence and the references above.
In my previous review, I also provided a pdf file describing 65 additional (minor) concerns and providing suggestions to solve most of them. Did the authors receive this file? I am attaching the file again in case the authors and the editor still find it useful.
We did miss this and have now made the suggested changes - see Word document with track changes uploaded. Except in two cases we did not make suggested edits:
- We did not change the section at lines 51-53 where we talk about genetic data because we dont think anything needs to change. The reviewer asks what does it tell us if we do not currently have genetic data- our answer is that it tells us that we should also follow up with genetic data if that is a direction we want to follow.
- We respectfully did not move the sentences around in the discussion suggested by the editor because we preferred it how we originally wrote it and think that makes the most sense.

Reviewer 2 Report
Thank you for addressing reviewer's comments.
Author Response
Thank you for reviewing our paper.